# QUICK-TUNE: QUICKLY LEARNING WHICH PRE-TRAINED MODEL TO FINETUNE AND HOW

**Sebastian Pineda Arango, Fabio Ferreira, Arlind Kadra, Frank Hutter & Josif Grabocka**
Department of Computer Science
University of Freiburg
`pineda@cs.uni-freiburg.de`

## ABSTRACT

With the ever-increasing number of pretrained models, machine learning practitioners are continuously faced with the decision of which pretrained model to use, and how to finetune it for a new dataset. In this paper, we propose a methodology that jointly searches for the optimal pretrained model and the hyperparameters for finetuning it. Our method transfers knowledge about the performance of many pretrained models with multiple hyperparameter configurations on a series of datasets. To this aim, we evaluated over 20k hyperparameter configurations for finetuning 24 pretrained image classification models on 87 datasets to generate a large-scale meta-dataset. We meta-learn a gray-box performance predictor on the learning curves of this meta-dataset and use it for fast hyperparameter optimization on new datasets. We empirically demonstrate that our resulting approach can quickly select an accurate pretrained model for a new dataset together with its optimal hyperparameters. To facilitate reproducibility, we open-source our code and release our meta-dataset.[1].

## 1 INTRODUCTION

Transfer learning has been a game-changer in the machine learning community, as finetuning pretrained deep models on a new task often requires much fewer data instances and less optimization time than training from scratch (Liu et al., 2021; You et al., 2020). Researchers and practitioners are constantly releasing pretrained models of different scales and types, making them accessible to the public through model hubs (a.k.a. model zoos or model portfolios) (Schürholt et al., 2022; Ramesh & Chaudhari, 2022). This raises a new challenge, as practitioners must select which pretrained model to use and how to set its hyperparameters (You et al., 2021b), but doing so via trial-and-error is time-consuming and suboptimal.

In this paper, we address the resulting problem of quickly identifying the optimal pretrained model for a new dataset and its optimal finetuning hyperparameters. Concretely, we present **Quick-Tune**, a Combined Algorithm Selection and Hyperparameter Optimization (CASH) (Thornton et al., 2013) technique for finetuning, which jointly searches for the optimal model and its hyperparameters in a Bayesian optimization setup. Our technical novelty is based on three primary pillars: *i) gray-box hyperparameter optimization (HPO)* for exploring learning curves partially by few epochs and effectively investing more time into the most promising ones, *ii) meta-learning* for transferring the information of previous evaluations on related tasks, and *iii) cost-awareness* for trading off time and performance when exploring the search space. By utilizing these three pillars, our approach can efficiently uncover top-performing Deep Learning pipelines (i.e., combinations of model and hyperparameters).

In summary, we make the following contributions:

- We present an effective methodology for quickly selecting models from hubs and jointly tuning their hyperparameters.

---

[1] `https://github.com/releaunifreiburg/QuickTune`

- We design an extensive search space that covers common finetuning strategies. In this space, we train and evaluate 20k model and dataset combinations to arrive at a large meta-dataset in order to meta-learn a gray-box performance predictor and benchmark our approach.

- We compare against multiple baselines, such as common finetuning strategies and state-of-the-art HPO methods, and show the efficacy of our approach by outperforming all of the competitor baselines.

## 2 RELATED WORK

**Finetuning Strategies**   Finetuning resumes the training on a new task from the pretrained weights. Even if the architecture is fixed, the user still needs to specify various details, such as learning rate and weight decay, because they are sensitive to the difference between the downstream and upstream tasks, or distribution shifts (Li et al., 2020; Lee et al., 2022). A common choice is to finetune only the top layers can improve performance, especially when the data is scarce (Yosinski et al., 2014). Nevertheless, recent work proposes to finetune the last layer only for some epochs and subsequently unfreeze the rest of the network (Chen et al., 2019a; Wang et al., 2023), to avoid the distortion of the pretrained information. To reduce overfitting, some techniques introduce different types of regularization that operate activation-wise (Kou et al., 2020; Li et al., 2020; Chen et al., 2019b), parameter-wise (Li et al., 2018), or directly using data from the upstream task while finetuning (You et al., 2020; Zhong et al., 2020). No previous work studies the problem of jointly selecting the model to finetune and its optimal hyperparameters. Moreover, there exists no consensus on what is the best strategy to use or whether many strategies should be considered jointly as part of a search space.

**Model Hubs**   It has been a common practice in the ML community to make large sets of pretrained models publicly available. They are often referred to as model hubs, zoos, or portfolios. In computer vision, in the advent of the success of large language models, a more recent trend is to release all-purpose models (Oquab et al., 2023; Radford et al., 2021; Kirillov et al., 2023) which aim to perform well in a broad range of computer vision tasks. Previous work has argued that a large pretrained model can be sufficient for many tasks and may only need little hyperparameter tuning (Kolesnikov et al., 2020). However, recent studies also show strong evidence that scaling the model size does not lead to a one-model-fits-all solution in computer vision (Abnar et al., 2022). Besides presenting more diversity and flexible model sizes for adapting to variable tasks and hardware, model hubs can be used for regularized finetuning (You et al., 2021a), learning hyper-networks for generating the weights (Schürholt et al., 2022), learning to ensemble different architectures (Shu et al., 2022), ensembling the weights of similar architectures (Wortsman et al., 2022b; Shu et al., 2021; Wortsman et al., 2022a), or selecting a suitable model from the pool (Cui et al., 2018; Bao et al., 2019; Tran et al., 2019a; Nguyen et al., 2020; You et al., 2021b; Bolya et al., 2021). Previous work using model hubs does not analyze the interactions between the used model(s) and the hyperparameters and how to set them efficiently.

**HPO, Transfer HPO, and Zero-Shot HPO**   Several methods for Hyperparameter Optimization (HPO) have been proposed ranging from simple random search (Bergstra & Bengio, 2012a) to fitting surrogate models of true response, such as Gaussian processes (Rasmussen & Williams, 2006), random forests (Hutter et al., 2011), neural networks (Springenberg et al., 2016), hybrid techniques (Snoek et al., 2015), and selecting configurations that optimize predefined acquisition functions (Wilson et al., 2018). There also exist multi-fidelity methods that further reduce the wall-clock time necessary to arrive at optimal configurations (Li et al., 2017; Falkner et al., 2018a; Awad et al., 2021a; Shala et al., 2023a; Kadra et al., 2023). Transfer HPO can leverage knowledge from previous experiments to yield a strong surrogate model with few observations on the target dataset (Wistuba & Grabocka, 2021a; Pineda Arango & Grabocka, 2023; Shala et al., 2023b; Salinas et al., 2020). Methods that use meta-features, i.e., dataset characteristics that can be either engineered (Feurer et al., 2015; Wistuba et al., 2016) or learned (Jomaa et al., 2021), have also been proposed to warm-start HPO. Zero-shot HPO has emerged as an efficient approach that does not require any observations of the response on the target dataset, e.g. approaches that are model-free and use the average performance of hyperparameter configurations over datasets (Wistuba et al., 2015) or approaches that meta-learn surrogate models with a ranking loss (Khazi et al., 2023; Winkelmolen et al., 2020; Öztürk et al., 2022). In contrast to previous work, we propose to not only use the final

performance of configurations but to learn a Gaussian Process-based to predict the performance of partial learning curves as formulated by gray-box HPO approaches (Hutter et al., 2019).

## 3 MOTIVATION

Before introducing our method, we want to remind the reader about the importance of searching for the optimal pretrained neural network from a pool of models. Our main premise is that *there is no silver bullet model that fits all the finetuning tasks*. To illustrate this fact, we computed the error rates of a group of 24 efficient models (detailed in Section 5.1) from the *timm* library (Wightman, 2019) on all the 26 datasets of the *Extended* split of MetaAlbum (Ullah et al., 2022) (details in Section 5). For every model, we use its best per-dataset hyperparameter configuration found by a comprehensive HPO. Figure 1 shows the ranks of the 24 models for the 26 datasets, demonstrating that there is very little regularity. In particular, there exists no single model that ranks optimally on all datasets, even if we optimize its hyperparameters for each dataset. Since there exists no silver bullet model, and considering that there is a large number of pretrained models available in recent hubs, then *how can we quickly select the best model for a new dataset?*

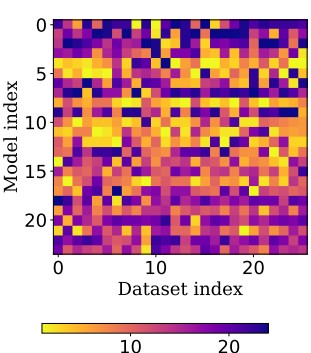

Figure 1: Ranks of model performances across datasets.

## 4 QUICK-TUNE: COST-EFFICIENT FINETUNING

Following our motivation, we aim to find the best pipeline $x = \{m, \lambda\}, x \in \mathcal{X}$, within a search space $\mathcal{X} := \mathcal{M} \times \Lambda$ comprising a model hub $m \in \mathcal{M}$ and a set of hyperparameters $\lambda \in \Lambda$. In this section, we detail how we solve this problem efficiently in order to yield competitive anytime performance.

### 4.1 QUICK-TUNE

We follow an efficient Bayesian Optimization strategy to search for the optimal pipelines, in a similar style to recent state-of-the-art approaches in HPO (Wistuba & Grabocka, 2021b; Wistuba et al., 2022). At every iteration, our method Quick-Tune fits estimators that predict the performance of pipelines and their cost (for details, see Section 4.2). Then it uses an acquisition function (detailed in Section 4.3) to select the next pipeline to continue finetuning for an incremental number of epochs. Finally, our method evaluates the loss and the runtime cost and adds it to the history. This procedure is repeated until a time budget is reached. We formalize these steps in Algorithm 1, where we use the validation loss as a performance metric. The entire procedure is sped up by starting from a meta-learned surrogate as described in Section 4.4.

---

**Algorithm 1:** Quick-Tune Algorithm

**Input:** Search space of pipelines $x \in \mathcal{X}$, Epoch step $\Delta t$
**Output:** Pipeline with the smallest observed loss
1  Select randomly a pipeline $x' \in \mathcal{X}$ and evaluate it for $\Delta t$ epochs ;
2  Initialize the history $\mathcal{H} \leftarrow \{(x', \Delta t, \ell(x', \Delta t), c(x', \Delta t))\}$
3  **while** $budget$ **do**
4     Update the performance predictor $\hat{\ell}$ from $\mathcal{H}$ using Equation 1;
5     Update the cost estimator $\hat{c}$ from $\mathcal{H}$ using Equation 2;
6     Select the next pipeline $x^*$ using Equation 3;
7     Evaluate the performance $\ell(x^*, \tau(x^*))$ and measure the cost $c(x^*, \tau(x^*))$ ;
8     Update the history $\mathcal{H} \leftarrow \mathcal{H} \cup \{(x^*, \tau(x^*), \ell(x^*, \tau(x^*)), c(x^*, \tau(x^*)))\}$ ;
9  **end**
10  **return** $\arg\min_{x \in \mathcal{X}} \{\ell(x, t) \mid (x, t, \ell(x, t), \cdot) \in \mathcal{H}\}$;

---

## 4.2 PERFORMANCE AND COST ESTIMATORS

Learning curves record the performance of Deep Learning pipelines at different time steps, such as the validation loss versus the number of epochs. The performance of the pipeline $x$ at step $t$ is denoted as $\ell(x,t)$, and the runtime cost for training the pipeline $x$ until step $t$ is $c(x,t)$. The history of all observed learning curves for $n$ pipelines is denoted as $\mathcal{H} := \{(x_i, t_i, \ell(x_i, t_i), c(x_i, t_i))\}_{i=1}^{n}$.

Our method learns a probabilistic performance estimator (a.k.a. surrogate) defined as $\hat{\ell}(x, t; \theta)$ and parameterized with $\theta$. We train the surrogate $\hat{\ell}$ to estimate the true performance $\ell$ from $\mathcal{H}$ as:

$$\theta^* := \arg\min_{\theta} \ \mathbb{E}_{(x,t,\ell(x,t),\cdot)\sim\mathcal{H}} \left[ -\log p\left(\ell(x,t) \mid x,t,\hat{\ell}(x,t;\theta)\right) \right]. \tag{1}$$

Concretely, the surrogate $\hat{\ell}$ is implemented as a deep-kernel Gaussian Process regressor (Wistuba & Grabocka, 2021a). In addition, we train a cost estimator $\hat{c}(x, t; \gamma)$ in the form of a Multilayer Perceptron with parameters $\gamma$ to predict the ground truth costs as:

$$\gamma^* := \arg\min_{\gamma} \ \mathbb{E}_{(x,t,\cdot,c(x,t))\sim\mathcal{H}} \left[ c(x,t) - \hat{c}(x,t;\gamma) \right]^2. \tag{2}$$

## 4.3 COST-SENSITIVE ACQUISITION FUNCTION

We propose a cost-sensitive variant of the Expected Improvement (Jones et al., 1998) (EI) acquisition to select the next pipeline to evaluate within a Bayesian Optimization framework, defined as:

$$x^* := \arg\max_{x\in\mathcal{X}} \frac{\text{EI}\left(x, \mathcal{H}, \hat{\ell}(x, \tau(x))\right)}{\hat{c}\left(x, \tau(x)\right) - c\left(x, \tau(x) - \Delta t\right)} = \arg\max_{x\in\mathcal{X}} \frac{\mathbb{E}_{\hat{\ell}(x,\tau(x))}\left[\max\left(\ell_{\tau(x)}^{\min} - \hat{\ell}(x,\tau(x)), 0\right)\right]}{\hat{c}\left(x, \tau(x)\right) - c\left(x, \tau(x) - \Delta t\right)} \tag{3}$$

The numerator of Equation 3 introduces a mechanism that selects the pipeline $x$ that has the largest likelihood to improve the lowest observed validation error at the next unobserved epoch $\tau(x)$ of pipeline $x$. The denominator balances out the cost of actually finetuning pipeline $x$ for $\Delta t$ epochs. $\tau(x)$ is defined for pipeline $x$ as $\tau(x) := \max\{t' | (x, t', \cdot, \cdot) \in \mathcal{H}\} + \Delta t$, where $\Delta t$ denotes the number of epochs to finetune from the last observed epoch in the history. If the pipeline is not in the history, the query epoch is $\tau(x) = \Delta t$. Simply put, if the validation loss of $x$ is evaluated after every training epoch/step ($\Delta t = 1$) and has been evaluated for $k$ epochs/steps, then $\tau(x) = k + 1$. As a result, we select the configuration with the highest chance of improving the best-measured loss at the next epoch, while trading off the cost of finetuning it. Concretely, the best observed loss is $\ell_{\tau(x)}^{\min} := \min\left(\{\ell(x,\tau(x)) | (x,\tau(x), \ell(x,\tau(x)), \cdot) \in \mathcal{H}\}\right)$. If no pipeline has been evaluated until $\tau(x)$, i.e. $(x,\tau(x),\cdot,\cdot) \notin \mathcal{H}$, then $\ell_{\tau(x)}^{\min} := \min\left(\{\ell(x,t) | (x,t,\ell(x,t),\cdot) \in \mathcal{H}, t < \tau(x)\}\right)$.

## 4.4 META-LEARNING THE PERFORMANCE AND COST ESTIMATORS

A crucial novelty of our paper is to meta-learn BO surrogates from existing pipeline evaluations on other datasets. Assume we have access to a set of curves for the validation errors $\ell$ and the runtimes $c$ of pipelines over a pool of datasets, for a series of $N$ epochs. We call the collection of such quadruple evaluations a meta-dataset $\mathcal{H}^{(M)} := \bigcup_{x\in\mathcal{X}} \bigcup_{d\in\mathcal{D}} \bigcup_{t\in[1,N]} \{(x,t,\ell(x,t,d),c(x,t,d))\}$, where we explicitly included the dependency of the performance and cost curves to the dataset. To contextualize the predictions on the characteristics of each dataset, we use descriptive features $d \in \mathcal{D}$ to represent each dataset (a.k.a. meta-features).

We meta-learn a probabilistic validation error estimator $\hat{\ell}(x,t,d;\theta)$, and a point-estimate cost predictor $\hat{c}(x,t,d;\gamma)$ from the meta-dataset $\mathcal{H}^{(M)}$ by solving the following objective functions:

$$\theta^{(M)} := \arg\min_{\theta} \mathbb{E}_{(x,t,\ell(x,t,d),c(x,t,d))\sim\mathcal{H}^{(M)}} \left[ -\log p \left( \ell(x,t,d) \mid x,t,d,\hat{\ell}(x,t,d;\theta) \right) \right] \quad (4)$$

$$\gamma^{(M)} := \arg\min_{\gamma} \mathbb{E}_{(x,t,\ell(x,t,d),c(x,t,d))\sim\mathcal{H}^{(M)}} \left( c(x,t,d) - \hat{c}(x,t,d;\gamma) \right)^2 \quad (5)$$

After meta-learning, we use the learned weights to initialize the performance and cost predictors $\theta \leftarrow \theta^{(M)}$ and $\gamma \leftarrow \gamma^{(M)}$ before running Algorithm 1. As a result, our method starts with a strong prior for the performance of pipelines and their runtime costs, based on the collected history $\mathcal{H}^{(M)}$ from evaluations on prior datasets. We provide details about the meta-learning procedure in Algorithm 2 (Appendix A.3).

## 5 QUICK-TUNE META-DATASET

### 5.1 QUICK-TUNE SEARCH SPACE

While our proposed method is agnostic to the application domain, the set of pretrained models and hyperparameter space to choose from, we need to instantiate these choices for our experiments. In this paper, we focus on image classification and base our study on the *timm* library (Wightman, 2019), given its popularity and wide adoption in the community. It contains a large set of hyperparameters and pretrained models on ImageNet (more than 700). Concerning the space of potential finetuning hyperparameters, we select a subset of optimizers and schedulers that are well-known and used by researchers and practitioners. We also include regularization techniques, such as data augmentation and drop-out, since finetuning is typically applied in low data regimes where large architectures easily overfit. Additionally, we modified the framework to include common finetuning

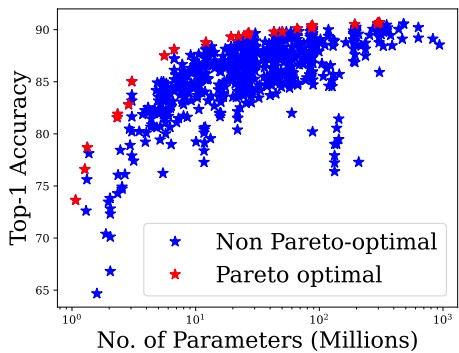

Figure 2: The subset of Pareto optimal pretrained models with respect to the predictive accuracy and model size.

strategies, such as methods to select the percentage of layers to finetune (Yosinski et al., 2014), linear probing (Wang et al., 2023), stochastic norm (Kou et al., 2020), Co-Tuning (You et al., 2020), DELTA (Li et al., 2019), BSS (Chen et al., 2019b) and SP-regularization (Li et al., 2018). The last five methods are taken from the *transfer learning library* (Junguang Jiang & Long, 2020). Although we consider these well-known and stable finetuning strategies, we foresee the widespread adoption of new approaches such as LoRA (Hu et al., 2021). They are complementary to our method and can be easily interpreted as an extension of the pipeline search space. We list all the hyperparameters of our search space in Table 1, indicating explicitly the conditional hyperparameters with a "*". For a more detailed description of our search space, including the hyperparameter ranges and dependencies, we point the reader to Table 7 of Appendix B. As we are interested in time efficiency and accuracy, we select the Pareto optimal models from the large set of ca. 700 pretrained architectures in the *timm* library. Specifically, given a model $m \in \mathcal{M}_{\text{Timm}}$ with Top-1 ImageNet accuracy $f_{\text{ImageNet}}(m)$ and $S(m)$ number of parameters, we build our final model hub based on the multi-objective optimization among the predictive accuracy and model size by solving Equation 6. Subsequently, we obtain a set of 24 Pareto-optimal models as shown in Figure 2 and listed in Table 8 of Appendix B.

$$\mathcal{M} = \left\{ m^* \mid m^* \in \arg\max_{m \in \mathcal{M}_{\text{Timm}}} \left[ f_{\text{ImageNet}}(m), -S(m) \right] \right\} \quad (6)$$

### 5.2 META-DATASET GENERATION

We created a large meta-dataset of evaluated learning curves based on the aforementioned search space. Overall, we finetuned the 24 Pareto-optimal pretrained models on 86 datasets for different hyperparameter configurations (details in Table 6, Appendix B.1). For every dataset, we sample hyperparameter configurations and models uniformly at random from the search space of Table 7.

Table 1: Search Space Summary.

| Hyperparameter Group | Hyperparameters |
|---|---|
| **Finetuning Strategies** | Percentage of the Model to Freeze, Layer Decay, Linear Probing, Stochastic Norm, SP-Regularization, DELTA Regularization, BSS Regularization, Co-Tuning |
| **Regularization Techniques** | MixUp, MixUp Probability*, CutMix, Drop-Out, Label Smoothing, Gradient Clipping |
| **Data Augmentation** | Data Augmentation Type (Trivial Augment, Random Augment, Auto-Augment), Auto-Augment Policy*, Number of operations*, Magnitude* |
| **Optimization** | Optimizer type (SGD, SGD+Momentum, Adam, AdamW, Adamp), Beta-s*, Momentum*, Learning Rate, Warm-up Learning Rate, Weight Decay, Batch Size |
| **Learning Rate Scheduling** | Scheduler Type (Cosine, Step, Multi-Step, Plateau), Patience*, Decay Rate*, Decay Epochs* |
| **Model** | 24 Models on the Pareto front (see Appendix 8) |

In our experiments, we use the tasks contained in the Meta-Album benchmark (Ullah et al., 2022) since it contains a diverse set of computer vision datasets. The benchmark is released in three variants with an increasing number of images per dataset: *micro*, *mini*, and *extended*. Concretely, *micro* has computer vision tasks with fewer classes and fewer images per class than *extended*. When generating the learning curves, we limited each run to 50 training epochs. As setting a limit is challenging when considering a pool of models and tasks with different sizes, we decided to constrain the finetuning procedure using a global time limit. The configurations trained on the tasks from *micro, mini, extended* are finetuned for 1, 4, and 16 hours respectively, using a single NVIDIA GeForce RTX 2080 Ti GPU per finetuning task, amounting to a total compute time of 32 GPU months. We summarize the main characteristics of our generated data in Table 6 in the Appendix.

## 6 EXPERIMENTS AND RESULTS

### 6.1 QUICK-TUNE PROTOCOL

While Quick-Tune finds the best-pretrained models and their hyperparameters, it also has hyperparameters of its own: the architecture, the optimizer for the predictors, and the acquisition function. Before running the experiments, we aimed to design a single setup that easily applies to all the tasks. Given that we meta-train the cost and the performance predictor, we split the tasks per Meta-Album version into five folds $\mathcal{D} = \{\mathcal{D}_1, ..., \mathcal{D}_5\}$ containing an equal number of tasks. When searching for a pipeline on datasets of a given fold $\mathcal{D}_i$, we consider one of the remaining folds for meta-validation and the remaining ones for meta-training. We used the meta-validation for early stopping when meta-training the predictors.

We tune the hyperparameters of Quick-Tune's architecture and the learning rate using the *mini* version's meta-validation folds. For the sake of computational efficiency, we apply the same discovered hyperparameters in the experiments involving the other Meta-Album versions. The chosen setup uses an MLP with 2 hidden layers and 32 neurons per layer, for both predictors. We use the Adam optimizer with a learning rate of $10^{-4}$ for fitting the estimators during the BO steps. We update their parameters for 100 epochs for every iteration from Algorithm 1. Further details on the set-up are specified in Appendix A.2. The inputs to the cost and performance estimators are the dataset metafeatures (Appendix A.4) and a pipeline encoding that concatenates a categorical embedding of the model $m$, an embedding of the observed curve $\tau(x)$ and the hyperparameters $\lambda$ (details in Appendix A.5). Finally, for the acquisition function, we use $\Delta t = 1$ epoch as in previous work (Wistuba et al., 2022), since this allows us to discard bad configurations quickly during finetuning.

Table 2: Performance comparison for Hypothesis 1. Normalized regret, ranks and standard deviations are calculated across all respective Meta-Album (Ullah et al., 2022) subset datasets.

| | Normalized Regret | | | Rank | | |
|---|---|---|---|---|---|---|
| | **Micro** | **Mini** | **Extended** | **Micro** | **Mini** | **Extended** |
| **BEiT+Default HP** | $0.229_{\pm 0.081}$ | $0.281_{\pm 0.108}$ | $0.225_{\pm 0.059}$ | $2.583_{\pm 0.829}$ | $2.611_{\pm 0.465}$ | $3.136_{\pm 0.215}$ |
| **XCiT+Default HP** | $0.223_{\pm 0.075}$ | $0.290_{\pm 0.107}$ | $0.199_{\pm 0.057}$ | $2.500_{\pm 0.751}$ | $2.694_{\pm 0.264}$ | $2.522_{\pm 0.344}$ |
| **DLA+Default HP** | $0.261_{\pm 0.074}$ | $0.325_{\pm 0.111}$ | $0.219_{\pm 0.076}$ | $3.062_{\pm 0.770}$ | $3.138_{\pm 0.248}$ | $2.977_{\pm 0.284}$ |
| **Quick-Tune** | $\mathbf{0.153_{\pm 0.054}}$ | $\mathbf{0.139_{\pm 0.112}}$ | $\mathbf{0.052_{\pm 0.031}}$ | $\mathbf{1.854_{\pm 1.281}}$ | $\mathbf{1.555_{\pm 0.531}}$ | $\mathbf{1.363_{\pm 0.376}}$ |

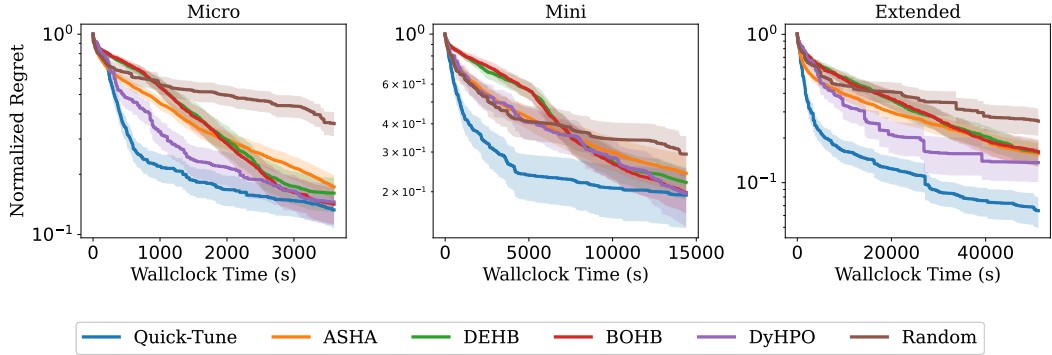

Figure 3: Comparison against state-of-the-art HPO methods.

## 6.2 RESEARCH HYPOTHESES AND ASSOCIATED EXPERIMENTS

**HYPOTHESIS 1**: QUICK-TUNE IS BETTER THAN FINETUNING MODELS WITHOUT HYPERPARAMETER OPTIMIZATION.

We argue that ML practitioners need to carefully tune the hyperparameters of pretrained models to obtain state-of-the-art performance. However, due to computational and time limitations, a common choice is to use default hyperparameters. To simulate the simplest practical use case, we select three different models from the subset of Pareto-optimal pretrained models (see Fig. 2), i.e. the largest model with the best accuracy (*beit_large_patch16_512* (Bao et al., 2022); 305M parameters, 90.69% acc.), the middle model with a competitive accuracy (*xcit_small_12_p8_384_dist* (Ali et al., 2021); 26M and 89.52%), as well as the smallest model with the lowest accuracy among the Pareto front models (*dla46x_c* (Yu et al., 2018); 1.3M and 72.61%). On each dataset in Meta-Album (Ullah et al., 2022), we finetune these models with their default hyperparameters and compare their performance against Quick-Tune. The default configuration is specified in Appendix B.2. To measure the performance, we calculate the average normalized regret (Arango et al., 2021), computed as detailed in Appendix A.1. For all Meta-Album datasets in a given category, we use the same finetuning time budget, i.e. 1 (*micro*), 4 (*mini*), and 16 (*extended*) hours. As reported in Table 2, Quick-Tune outperforms the default setups in terms of both normalized regret and rank across all respective subset datasets, demonstrating that HPO tuning is not only important to obtain high performance, but also achievable in low time budget conditions.

**HYPOTHESIS 2**: QUICK-TUNE OUTPERFORMS STATE-OF-THE-ART HPO OPTIMIZERS.

Gray-box approaches are considered very practical, especially for optimizing expensive architectures. We compare Quick-Tune against four popular gray-box optimizers, ASHA (Li et al., 2018), BOHB (Falkner et al., 2018b), DEHB (Awad et al., 2021b) and DyHPO (Wistuba et al., 2022). We additionally include Random Search (Bergstra & Bengio, 2012b) as a baseline for a sanity check. The normalized regret is computed for the three Meta-album versions on budgets of 1, 4 and 16 hours. The results of Figure 3 show that our proposed method has the best any-time performance compared to the baselines. In an additional experiment, presented in Figure 4, we show that both meta-training and cost-awareness aspects contribute to this goal by ablating each individual component. This

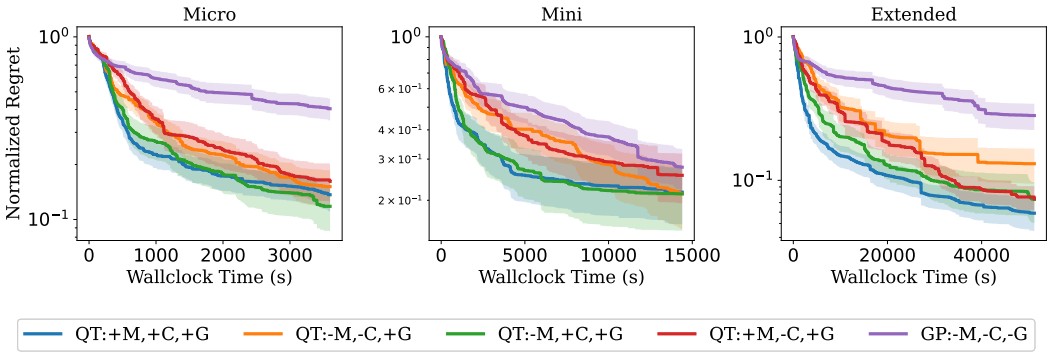

Figure 4: Comparing Quick-Tune with (+) and without (-) (M)eta-learning and (C)ost-Awareness, and (G)ray-box optimization. We also compare against DyHPO (=QT:-M,-C,+G) and a GP.

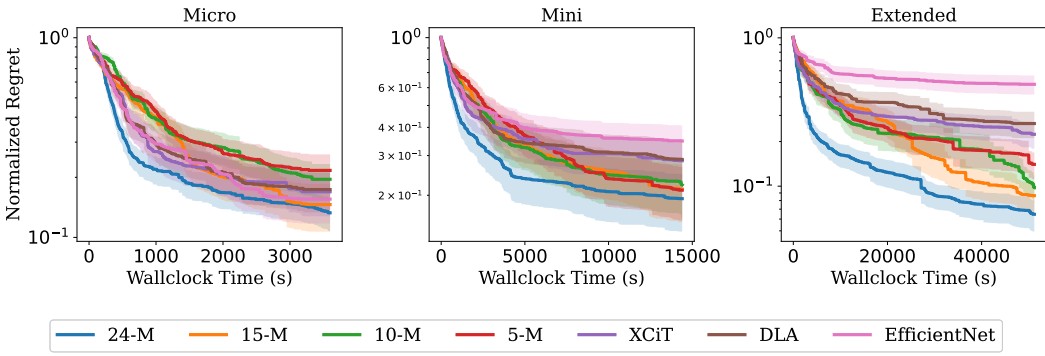

Figure 5: Varying the model hub size.

behavior is consistent among datasets of different sizes and present in all three meta-dataset versions. We attribute the search efficiency to our careful search space design, which includes both large and small models, as well as regularization techniques that reduce overfitting in low-data settings such as in the tasks of the *micro* version. In large datasets, our method finds good configurations even faster compared to the baselines, highlighting the importance of cost-awareness in optimizing hyperparameters for large datasets. We additionally compare against a Gaussian Process (GP) that observes the whole learning curve ($\Delta t = 50$), to highlight the necessity of a gray-box approach. In an additional experiment in Appendix C, we evaluate our method on the well-known Inaturalist (Horn et al., 2021) and Imagenette (Howard, 2019) datasets that are not contained in Meta-Album; there, our method still consistently outperforms the competitor baselines.

**HYPOTHESIS 3**: CASH ON DIVERSE MODEL HUBS IS BETTER THAN HPO ON A SINGLE MODEL.

A reasonable question is whether we actually need to consider a hub of models at all, or whether perhaps using a single, expressive, and well-tuned architecture is sufficient for most datasets. We hypothesize that the optimal model is dataset-specific because the complexities of datasets vary. Therefore, using a single model for all the datasets is a sub-optimal practice, and it is better to include a diverse model hub. Moreover, using a model hub allows us to explore cheap models first and gain information about the interactions between the hyperparameters. The information can in turn be leveraged by the predictors when considering larger and more accurate models.

To validate our hypothesis, we select *EfficientNet* (Tan & Le, 2019), *X-Cit* (Ali et al., 2021) and *DLA* (Yu et al., 2018). These correspond to models with at least 10 evaluated curves in all the datasets and are located on the top, middle, and bottom regions in the Pareto front. Subsequently, we optimize their hyperparameters independently using our Quick-Tune algorithm. We also run Quick-Tune on subsets of 5, 10, and 15 models out of the model hub $\mathcal{M}$ with 24 models. The subset of models was created randomly for every dataset before running BO. We execute the optimization on the three

Table 3: Comparison against efficient-finetuning of a single large model.

| | 4 Hours | | | 24 Hours | | |
| --- | --- | --- | --- | --- | --- | --- |
| | **Micro** | **Mini** | **Extended** | **Micro** | **Mini** | **Extended** |
| **Dinov2 + LoRA** | $0.541_{\pm 0.093}$ | $0.049_{\pm 0.018}$ | $0.055_{\pm 0.004}$ | $0.332_{\pm 0.095}$ | $0.014_{\pm 0.021}$ | $0.004_{\pm 0.012}$ |
| **Dinov2 + Linear Probing** | $0.081_{\pm 0.041}$ | $0.067_{\pm 0.021}$ | $0.081_{\pm 0.012}$ | $0.067_{\pm 0.038}$ | $0.017_{\pm 0.019}$ | $0.042_{\pm 0.011}$ |
| **QuickTune** | $\mathbf{0.072_{\pm 0.024}}$ | $\mathbf{0.039_{\pm 0.014}}$ | $\mathbf{0.042_{\pm 0.016}}$ | $\mathbf{0.018_{\pm 0.012}}$ | $\mathbf{0.012_{\pm 0.008}}$ | $\mathbf{0.003_{\pm 0.008}}$ |

Figure 6: Comparison with a two-stage search for models and hyperparameters.

meta-dataset versions for 1, 2 and 4 hours of total budget. Figure 5 demonstrates that, in general, it is better to have a pool of diverse models such as 24 models (24-M) or 15 models (15-M), than tuning a small set of models or even a unique model. Interestingly, we note the larger the dataset is, the larger the model hub we need.

**Quick-Tune vs. Efficient Finetuning of a Single Large Model.** Although we propose to use model hubs, practitioners also have the alternative of choosing a large pretrained model from outside or inside the hub. We argue that a large pretrained model still demands HPO (Oquab et al., 2023), and imposes a higher load on computing capabilities. To demonstrate that Quick-Tune still outperforms the aforementioned approach, we compare our method against efficient finetuning approaches of Dinov2 which features 1B parameters by; *i)* finetuning only the last layer of Dino v2, which represents a common practice in the community, and *ii)* finetuning with LoRA (Hu et al., 2021), a parameter-efficient finetuning method. Our results in Table 3 demonstrate that CASH on model hubs via QuickTune attains better results for the different dataset versions.

**Quick-Tune vs. Separated Model and Hyperparameter Optimization.** We compare Quick-Tune with a two-stage alternative approach where, we first select a model with its default hyperparameters using state-of-the-art model selection methods, such as LogME (You et al., 2021a), LEEP (Nguyen et al., 2020) and NCE (Tran et al., 2019b). Then, we conduct a second search for the optimal hyperparameters of the model selected in the first stage. The results reported in Figure 6 show that Quick-Tune outperforms this two-stage approach, thus highlighting the importance of performing combined HPO and model selection.

# 7 CONCLUSION

We tackle the practical problem of selecting a model and its hyperparameters given a pool of models. Our method QuickTune leverages gray-box optimization together with meta-learned cost and performance predictors in a Bayesian optimization setup. We demonstrate that QuickTune outperforms common strategies for selecting pretrained models, such as using single models, large feature extractors, or conventional HPO tuning methods. In addition, we present empirical evidence that our method outperforms large-scale and state-of-the-art transformer backbones for computer vision. As a consequence, QuickTune offers a practical and efficient alternative for selecting and tuning pretrained models for image classification.

## 8 ACKNOWLEDGEMENT

Robert Bosch GmbH is acknowledged for financial support. We also acknowledge funding by the Deutsche Forschungsgemeinschaft (DFG, German Research Foundation) under SFB 1597 (Small-Data), grant number 499552394, the support of the BrainLinks- BrainTools Center of Excellence, and the funding of the Carl Zeiss foundation through the ReScaLe project. This research was also partially supported by the Deutsche Forschungsgemeinschaft (DFG, German Research Foundation) under grant number 417962828, by the state of Baden-Württemberg through bwHPC, and the German Research Foundation (DFG) through grant no INST 39/963-1 FUGG, by TAILOR, a project funded by the EU Horizon 2020 research, and innovation program under GA No 952215, and by European Research Council (ERC) Consolidator Grant "Deep Learning 2.0" (grant no. 101045765). Funded by the European Union. Views and opinions expressed are however those of the authors only and do not necessarily reflect those of the European Union or the ERC. Neither the European Union nor the ERC can be held responsible for them.

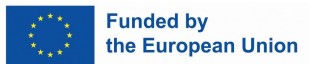

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

# A  ALGORITHMIC DETAILS

## A.1  NORMALIZED REGRET

Given an observed performance $y$, the normalized regret is computed per datasets as follows:

$$y_{\text{norm}} = \frac{y_{\max} - y}{y_{\max} - y_{\min}} \tag{7}$$

where $y_{\max}, y_{\min}$ in Equation 7 are respectively the maximum and minimum performances in the meta-dataset.

## A.2  ADDITIONAL SET-UP DETAILS

The categorical encoder of the model is a linear layer with 4 output neurons, while the learning curve embedding is generated from a convolutional neural network with two layers. For the rest of the hyperparameters of the deep-kernel Gaussian process surrogate and the acquisition function, we followed the settings described in the respective publication (Wistuba et al., 2022) and always used the setup suggested from the authors unless specified otherwise. We use the Synetune library (Salinas et al., 2022) for the implementation of the baselines.

## A.3  META-TRAINING ALGORITHMS

We present the procedure for meta-training the cost and loss predictors in Algorithm 2. Initially, we sample random batches after choosing a dataset randomly within our meta-dataset, and then we update the parameters of each predictor so that it minimizes their respective losses. The same strategy is used when updating during BO but with fewer iterations. We meta-train for 10000 iterations using the Adam Optimizer with a learning rate of 0.0001.

---

**Algorithm 2:** Meta-training Algorithm

**Input:** Metadata with precomputed losses and cost $\mathcal{H}^{(M)}$ with datasets $\mathcal{D} = \{d_1, ..., d_N\}$,
      learning rate $\mu$, Epochs $E$
**Output:** Meta-learned parameters $\theta, \gamma$

1  Random initialize parameters $\theta, \gamma$ for loss predictor $\hat{\ell}(\cdot)$ and cost predictor $\hat{c}(\cdot)$;
2  **for** $i \in \{1...E\}$ **do**
3      Sample dataset index $i \sim U[1, |\mathcal{D}|]$ and its metafeatures $d_i$;
4      Define the subset of history associated to $d_i$: $\mathcal{H}_i \subset \mathcal{H}^{(M)} : \{(x, t, \ell(x, t, d_i), c(x, t, d_i))\}$ ;
5      Compute $\delta\theta = -\nabla_\theta \sum_{(x,t,\ell(x,t,d_i),\cdot)\sim\mathcal{H}_i} \left[ \log p\left( \ell(x,t,d_i) \mid x,t,d,\hat{\ell}(x,t,d_i;\theta) \right) \right]$ ;
6      Compute $\delta\gamma = \nabla_\gamma \sum_{(x,t,\cdot,c(x,t,d_i))\sim\mathcal{H}_i} \left[ c(x,t,d_i) - \hat{c}(x,t,d_i;\gamma) \right]^2$ ;
7      Update parameters $\theta = \theta - \mu \cdot \delta\theta$, $\gamma = \gamma - \mu \cdot \delta\gamma$
8  **end**
9  $\theta^{(M)} \leftarrow \theta, \gamma^{(M)} \leftarrow \gamma$;
10 **return** $\theta^{(M)}, \gamma^{(M)}$

---

## A.4  META-FEATURES

Similar to previous work (Öztürk et al., 2022), we use descriptive meta-features of the dataset: number of samples, image resolution, number of channels, and number of classes. Any other technique for embedding datasets is compatible and orthogonal with our approach.

## A.5  PIPELINE ENCODING

Our pipeline encoding is the concatenation of the hyperparameters $\lambda_i$, the embedding of the model name $\mathcal{E}_{\text{model}}(m_i)$, and the embedding of the learning curves. Given the performance curve $\tau(x_i, t)$,

we obtain the respective embedding $\mathcal{E}_{\text{perf}}(\tau(x_i, t))$ using a 2-layer convolutional networks following a similar setup from previous work (Wistuba et al., 2022). For obtaining the model embedding, we transform the model names into one-hot-encoding and feed this representation into a linear layer (Pineda Arango & Grabocka, 2023). The pipeline encoding is finally defined as:

$$\text{Pipeline Encodig}(x_i) = [\lambda_i, \mathcal{E}_{\text{model}}(m_i), \mathcal{E}_{\text{perf}}(\tau(x, t))] \tag{8}$$

The parameters of the encoders $\mathcal{E}_{\text{model}}(\cdot), \mathcal{E}_{\text{perf}}(\cdot)$ are jointly updated during meta-training and while fitting the predictors during BO.

## B  META-DATASET DETAILS

### B.1  META-DATASET COMPOSITION DETAILS

While generating the meta-dataset, we take into account the dependencies of the conditional hyperparameters. Every triple (model, dataset, hyperparameter configuration) resulted in a finetuning optimization run that produced a validation error and cost curves. A few of the combinations are infeasible to evaluate due to the model size, thus some triples can have fewer evaluations. For instance, some pipelines failed due to the GPU requirements demanded by the number of parameters of the model and the number of classes of the datasets. In that case, we decreased the batch size iteratively, halving the value, until it fits to the GPU. In some cases, this strategy was not enough, thus some models have more evaluations than others. In Table 4, we present the list of datasets per set and indicate the heavy datasets with a (*), i.e. with a lot of classes or a lot of samples in the extended version. The majority of the datasets are present in all three versions of Meta-Album, except the underlined ones, which are not present in the extended version. The OpenML Ids associated to the datasets are listed in Table 5. Table 6 provides descriptive statistics regarding the generated meta-dataset for every corresponding Meta-Album version.

Table 4: Datasets per Set in Meta-Album

| Set | Dataset Names |
|-----|---------------|
| 0 | BCT, BRD*, CRS, FLW, MD_MIX, PLK*, PLT_VIL*, RESISC, SPT, TEX |
| 1 | ACT_40, APL, DOG, INS_2*, MD_5_BIS, MED_LF, PLT_NET*, PNU, RSICB, TEX_DTD |
| 2 | ACT_410, AWA*, BTS*, FNG, INS*, MD_6, PLT_DOC, PRT, RSD*, TEX_ALOT* |

Table 5: OpenML IDs for Datasets per Split and Version

| Version | Set 0 | Set 1 | Set 2 |
|---------|-------|-------|-------|
| Micro | 44241, 44238, 44239, 44242, 44237, 44246, 44245, 44244, 44240, 44243 | 44313, 44248, 44249, 44314, 44312, 44315, 44251, 44250, 44247, 44252 | 44275, 44276, 44272, 44273, 44278, 44277, 44279, 44274, 44271, 44280 |
| Mini | 44285, 44282, 44283, 44286, 44281, 44290, 44289, 44288, 44284, 44287 | 44298, 44292, 44293, 44299, 44297, 44300, 44295, 44294, 44291, 44296 | 44305, 44306, 44302, 44303, 44308, 44307, 44309, 44304, 44301, 44310 |
| Extended | 44320, 44317, 44318, 44321, 44316, 44324, 44323, 44322, 44319 | 44331, 44326, 44327, 44332, 44330, 44333, 44329, 44328, 44325 | 44338, 44340, 44335, 44336, 44342, 44341, 44343, 44337, 44334 |

Table 6: Quick-Tune Composition

| Meta-Dataset | Number of Tasks | Number of Curves | Total Epochs | Total Run Time |
|--------------|-----------------|------------------|--------------|----------------|
| Micro | 30 | 8.712 | 371.538 | 2.076 GPU Hours |
| Mini | 30 | 6.731 | 266.384 | 6.049 GPU Hours |
| Extended | 26 | 4.665 | 105.722 | 15.866 GPU Hours |

### B.2 HYPERPARAMETER SEARCH SPACE

Table 7: Detailed Search Space for Curve Generation. Bold font indicates the default configuration.

| Hyperparameter Group | Name | Options | Conditional |
|---|---|---|---|
| **Fine-Tuning Strategies** | Percentage to freeze | **0**, 0.2, 0.4, 0.6, 0.8, 1 | |
| | Layer Decay | **None**, 0.65, 0.75 | - |
| | Linear Probing | True, **False** | - |
| | Stochastic Norm | True, **False** | - |
| | SP-Regularization | **0**, 0.0001, 0.001, 0.01, 0.1 | - |
| | DELTA Regularization | **0**, 0.0001, 0.001, 0.01, 0.1 | - |
| | BSS Regularization | **0**, 0.0001, 0.001, 0.01, 0.1 | - |
| | Co-Tuning | **0**, 0.5, 1, 2, 4 | - |
| **Regularization Techniques** | MixUp | **0**, 0.2, 0.4, 1, 2, 4, 8 | |
| | MixUp Probability | **0**, 0.25, 0.5, 0.75, 1 | - |
| | CutMix | **0**, 0.1, 0.25, 0.5, 1,2,4 | - |
| | DropOut | **0**, 0.1, 0.2, 0.3, 0.4 | - |
| | Label Smoothing | **0**, 0.05, 0.1 | - |
| | Gradient Clipping | **None**, 1, 10 | - |
| **Data Augmentation** | Data Augmentation | **None**, trivial_augment, random_augment, auto_augment | - |
| | Auto Augment | None, v0, original | - |
| | Number of Operations | 2,3 | Data Augmentation (Random Augment) |
| | Magnitude | 9, 17 | Data Augmentation (Random Augment) |
| **Optimizer Related** | Optimizer Type | **SGD**, SGD+Momentum, Adam, AdamW, Adamp | - |
| | Betas | (0.9, 0.999), (0, 0.99), (0.9, 0.99), (0, 0.999) | Scheduler Type (Adam, Adamw, Adamp) |
| | Learning Rate | **0.1**,0.01, 0.005, 0.001, 0.0005, 0.0001, 0.00005, 0.00001 | - |
| | Warm-Up Learning Rate | **0**, 0.000001, 0.00001 | - |
| | Weight Decay | **0**, 0.00001, 0.0001, 0.001, 0.01,0.1 | - |
| | Batch Size | 2,4,8,16,32,64,**128**,256,512 | - |
| | Momeutm | **0**, 0.8, 0.9, 0.95, 0.99 | Optimizer Type (SGD+Momentum) |
| **Scheduler Related** | Scheduler Type | None, **Cosine**, Step, Multistep, Plateau | - |
| | Patience | 2,5 | Scheduler Type (Plateau) |
| | Decay Rate | 0.1, 0.5 | Scheduler Type (Step, Multistep) |
| | Decay Epochs | 10, 20 | Scheduler Type (Step, Multistep) |
| **Model** | Model | See Table 8 | |

Table 7 shows the complete search space of hyperparameters. During the curve generation, we sample uniformly among these discrete values. Some hyperparameters are conditional, i.e. their are only

present when another hyperparameter gets a specific set of values. Thus, we also list explicitly which are the conditions for such hyperparameters.

We report the hyperparameters for the default configuration in Experiment 1 by using a bold font in Table 7.

### B.3    MODEL HUB

We list all the models on the Pareto Front from Timm's library as provided on version *0.7.0dev0*. Moreover, we report their size (number of parameters) and the top-1 accuracy in ImageNet.

Table 8: Models on the pareto front

| Model Name | No. of Param. | Top-1 Acc. |
| --- | --- | --- |
| beit_large_patch16_512 | 305.67 | 90.691 |
| volo_d5_512 | 296.09 | 90.610 |
| volo_d5_448 | 295.91 | 90.584 |
| volo_d4_448 | 193.41 | 90.507 |
| swinv2_base_window12to24_192to384_22kft1k | 87.92 | 90.401 |
| beit_base_patch16_384 | 86.74 | 90.371 |
| volo_d3_448 | 86.63 | 90.168 |
| tf_efficientnet_b7_ns | 66.35 | 90.093 |
| convnext_small_384_in22ft1k | 50.22 | 89.803 |
| tf_efficientnet_b6_ns | 43.04 | 89.784 |
| volo_d1_384 | 26.78 | 89.698 |
| xcit_small_12_p8_384_dist | 26.21 | 89.515 |
| deit3_small_patch16_384_in21ft1k | 22.21 | 89.367 |
| tf_efficientnet_b4_ns | 19.34 | 89.303 |
| xcit_tiny_24_p8_384_dist | 12.11 | 88.778 |
| xcit_tiny_12_p8_384_dist | 6.71 | 88.101 |
| edgenext_small | 5.59 | 87.504 |
| xcit_nano_12_p8_384_dist | 3.05 | 85.025 |
| mobilevitv2_075 | 2.87 | 82.806 |
| edgenext_x_small | 2.34 | 81.897 |
| mobilevit_xs | 2.32 | 81.574 |
| edgenext_xx_small | 1.33 | 78.698 |
| mobilevit_xxs | 1.27 | 76.602 |
| dla46x_c | 1.07 | 73.632 |

## C    ADDITIONAL EXPERIMENT: QUICK-TUNE ON DATASETS OUTSIDE META-ALBUM

Meta-album contains a broad set of datasets, ranging from small-size datasets with 20 samples per class and 20 classes, to more than 700 classes with up to 1000 samples per class. Moreover, it offers a diverse set of domains that foster a strong benchmarking of image classification methods. To further verify the generalization outside the curated datasets present in Meta-Album, we run experiments on two well-known datasets that are not present in Meta-Album. Initially, we run Quick-Tune on Imagenette (Howard, 2019) by using a time budget of 4 hours. Additionally, we run Quick-Tune it on Inaturalist (Horn et al., 2021) with a time budget of 16 hours. Finally, we transfer the estimators meta-learned on the *mini* and *extended* splits respectively. We compare the results to the same gray-box HPO baselines as Hypothesis 2. The selection of the budget and the transferred estimators is based on the similarity of each dataset size to the corresponding Meta-Album super-set.

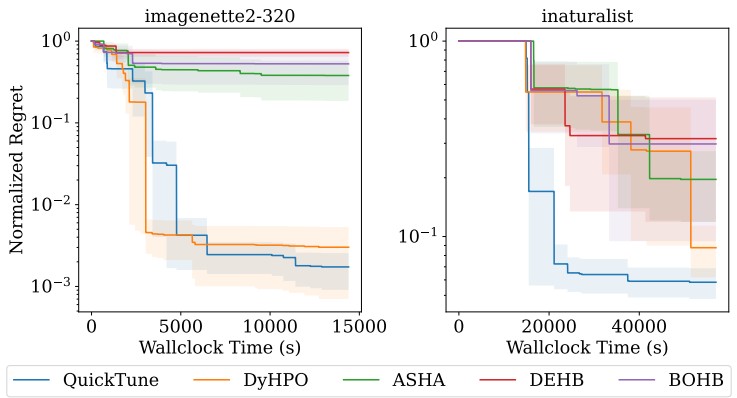

Figure 7: Evaluation of Quick-Tune on Datasets outside Meta-Album.

# D    DETAILS ON DINOV2 EXPERIMENT

## D.1    FINETUNING LAST LAYER IN DINOV2

A common practice is to use large feature extractors as the backbone and just train a linear output layer. We argue that selecting the model from a pool and optimizing its hyperparameters jointly is a more effective approach. Firstly, large backbones are often all-purpose models that may be inferior to model hubs when downstream tasks deviate largely from the backbone pretraining and may require non-trivial finetuning hyperparameter adaptations. As such, individual large models may violate the diversity property observed in our third hypothesis above. Secondly, due to their large number of parameters, they are expensive to optimize.

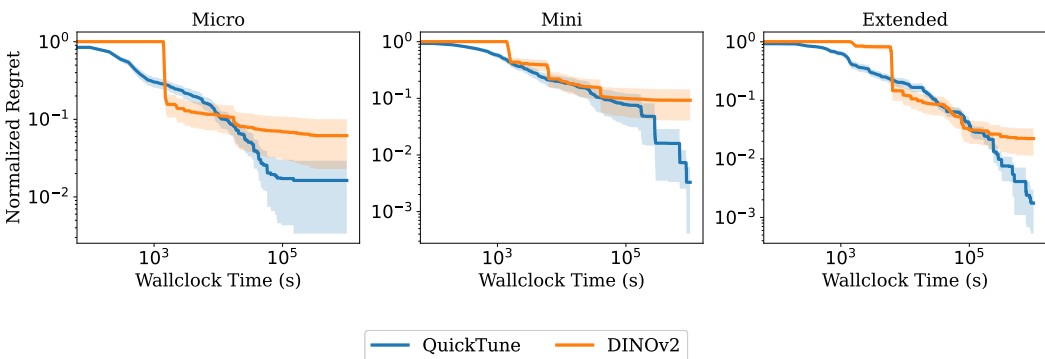

Figure 8: Results for finetuning the last layer of DINOv2. We relax the efficiency conditions by allowing a bigger time budget but still limited to 1 GPU.

To verify that selecting a model from a model hub with Quick-Tune is a more effective approach than using an all-purpose model, we compare to DINOv2 (Oquab et al., 2023) According to the method's linear evaluation protocol, the procedure to classify downstream tasks with pretrained DINOv2 involves performing a small grid search over a subset of its finetuning hyperparameters (104 configurations in total including learning rate, number of feature extraction layers, etc.). We adopt this grid search in our comparison, evaluating all the hyperparameter configurations on the grid. For each meta-album version, we then compare the normalized regret against the wall-clock time between DINOv2 and Quick-Tune. For the experiment, we increase Quick-Tune's budget to match DINOv2's budget requirements since the evaluation of the full grid of DINOv2 requires more time than our previous experiments. In Figure 8, we present the results of our comparison, where, our method manages to outperform DINOv2 in all the considered benchmarks, highlighting the benefits of our designed search space.

While performing our comparison, a small minority of the DINOv2 runs failed due to GPU memory limitations, and for a few runs, we had to minimally adjust the DINOv2 default hyperparameter configurations "$n\_last\_blocks$" to adapt to our GPU memory limitation. Tables 9, 10, 11 depict for which of the datasets we ran with the default hyperparameter configurations according to (Oquab et al., 2023) and which we adapted due to the single-GPU constraint (RTX2080). Runs indicated with "*" failed due to GPU memory limitations and for runs indicated by "$n\_last\_blocks = 1$" we ran with the default hyperparameters except for the "n_last_blocks" argument that had to be changed from 4 to 1 to fit on the GPU.

Table 9: Subset Micro

| Dataset | Linear Eval. Hyp. |
|---|---|
| micro_set0_BCT | DINOv2 default |
| micro_set0_BRD | DINOv2 default |
| micro_set0_CRS | DINOv2 default |
| micro_set0_FLW | DINOv2 default |
| micro_set0_MD_MIX | DINOv2 default |
| micro_set0_PLK | DINOv2 default |
| micro_set0_PLT_VIL | DINOv2 default |
| micro_set0_RESISC | DINOv2 default |
| micro_set0_SPT | DINOv2 default |
| micro_set0_TEX | DINOv2 default |
| micro_set1_ACT_40 | DINOv2 default |
| micro_set1_APL | DINOv2 default |
| micro_set1_DOG | DINOv2 default |
| micro_set1_INS_2 | DINOv2 default |
| micro_set1_MD_5_BIS | DINOv2 default |
| micro_set1_MED_LF | DINOv2 default |
| micro_set1_PLT_NET | DINOv2 default |
| micro_set1_PNU | DINOv2 default |
| micro_set1_RSICB | DINOv2 default |
| micro_set1_TEX_DTD | DINOv2 default |
| micro_set2_ACT_410 | DINOv2 default |
| micro_set2_AWA | DINOv2 default |
| micro_set2_BTS | DINOv2 default |
| micro_set2_FNG | DINOv2 default |
| micro_set2_INS | DINOv2 default |
| micro_set2_MD_6 | DINOv2 default |
| micro_set2_PLT_DOC | DINOv2 default |
| micro_set2_PRT | DINOv2 default |
| micro_set2_RSD | DINOv2 default |
| micro_set2_TEX_ALOT | DINOv2 default |

Table 10: Subset Mini

| Dataset | Linear Eval. Hyp. |
|---|---|
| mini_set0_BCT | DINOv2 default |
| mini_set0_BRD | n_last_blocks=1 |
| mini_set0_CRS | n_last_blocks=1 |
| mini_set0_FLW | DINOv2 default |
| mini_set0_MD_MIX | n_last_blocks=1 |
| mini_set0_PLK | DINOv2 default |
| mini_set0_PLT_VIL | DINOv2 default |
| mini_set0_RESISC | DINOv2 default |
| mini_set0_SPT | DINOv2 default |
| mini_set0_TEX | DINOv2 default |
| mini_set1_ACT_40 | DINOv2 default |
| mini_set1_APL | DINOv2 default |
| mini_set1_DOG | n_last_blocks=1 |
| mini_set1_INS_2 | n_last_blocks=1 |
| mini_set1_MD_5_BIS | n_last_blocks=1 |
| mini_set1_MED_LF | DINOv2 default |
| mini_set1_PLT_NET | DINOv2 default |
| mini_set1_PNU | DINOv2 default |
| mini_set1_RSICB | DINOv2 default |
| mini_set1_TEX_DTD | DINOv2 default |
| mini_set2_ACT_410 | DINOv2 default |
| mini_set2_AWA | DINOv2 default |
| mini_set2_BTS | DINOv2 default |
| mini_set2_FNG | DINOv2 default |
| mini_set2_INS | n_last_blocks=1 |
| mini_set2_MD_6 | n_last_blocks=1 |
| mini_set2_PLT_DOC | DINOv2 default |
| mini_set2_PRT | DINOv2 default |
| mini_set2_RSD | DINOv2 default |
| mini_set2_TEX_ALOT | n_last_blocks=1 |

Table 11: Subset Extended

| Dataset | Linear Eval. Hyp. |
|---|---|
| extended_set0_BCT | n_last_blocks=1 |
| extended_set0_CRS | n_last_blocks=1 |
| extended_set0_FLW | n_last_blocks=1 |
| extended_set0_RESISC | n_last_blocks=1 |
| extended_set0_SPT | n_last_blocks=1 |
| extended_set0_TEX | n_last_blocks=1 |
| extended_set1_ACT_40 | DINOv2 default |
| extended_set1_APL | n_last_blocks=1 |
| extended_set1_DOG | n_last_blocks=1 |
| extended_set2_ACT_410 | DINOv2 default |
| extended_set2_PLT_DOC | DINOv2 default |
| extended_set0_BRD | * |
| extended_set0_PLK | * |
| extended_set0_PLT_VIL | * |
| extended_set1_INS_2 | * |
| extended_set1_MED_LF | n_last_blocks=1 |
| extended_set1_PLT_NET | * |
| extended_set1_PNU | n_last_blocks=1 |
| extended_set1_RSICB | * |
| extended_set1_TEX_DTD | n_last_blocks=1 |
| extended_set2_AWA | * |
| extended_set2_BTS | * |
| extended_set2_FNG | n_last_blocks=1 |
| extended_set2_PRT | n_last_blocks=1 |
| extended_set2_RSD | * |
| extended_set2_TEX_ALOT | n_last_blocks=1 |
| extended_set2_INS | * |

## D.2 DINOV2 EFFICIENT FINETUNING WITH LORA

As an alternative, instead of the infeasible full finetuning, we finetune DINOv2 with LoRA [2], which is a state-of-the-art method for finetuning transformer models (such as DINOv2). LoRA demands ca. $1\%$ of the parameters of the full finetuning strategy and fits into our GPU restrictions for most of the datasets. Furthermore, LoRA is reported to achieve similar or better performance compared to full finetuning [2].

We stress that even with LoRA, the large DINOv2 model (1B params) exceeds our GPU memory capacity in some datasets. Thus, we present the results of the experiments in Table 3 with the datasets that DINOv2 were successfully trained, namely: 30 datasets for micro, 20 datasets for mini, and 4 datasets for extended. We report results for 4 Hours (4H) and 24 Hours (24H) of total budget where QuickTune outperforms both alternatives of finetuning DINOv2.

