# OpenReview forum: "Quick-Tune: Quickly Learning Which Pretrained Model to Finetune and How"
_ICLR.cc/2024/Conference — ICLR 2024 oral_

### Official Review · Reviewer_omMr · 2023-10-29

**Soundness:** 3 good
**Presentation:** 4 excellent
**Contribution:** 3 good
**Rating:** 8
**Confidence:** 4

**Summary:**

Currently there are many models available online but it is challenging to decide which of them to select and how to fine-tune it to the downstream task. The paper focuses on this challenge and introduces a method to find a suitable pretrained model together with suitable hyperparameters for fine-tuning. The authors construct a meta-dataset and use it for meta-learning a performance predictor that can jointly select the model and the hyperparameters.

**Strengths:**

* The paper studies an interesting problem setting that is practically important and likely to further increase in importance. The trend is that there will be more and more pretrained models available and these will be more commonly used, so a method for deciding how to select a suitable one and fine-tune it is valuable.
* The method seems to be novel and appears to be a well-designed solution to the presented problem. It also offers strong performance compared to the baselines that are evaluated.
* The paper is well-written and easy to read. The figures and tables are well-presented and make it simpler to understand the work and the results. The questions that are studied are clearly stated.
* The paper contributes a meta-dataset that can be interesting for other researchers to study this important problem.
* A variety of relevant research questions are asked and answered reasonably well.

**Weaknesses:**

* The paper only considers computer vision classification, while related works such as LogMe (You et al., ICML’21) consider various downstream tasks (classification and regression), and modalities (vision and language). It can be impractically expensive though to evaluate the approach on also these other settings.

**Questions:**

* There are micro, mini and extended versions of the Meta-Album datasets, which poses the question if Quick-Tune could benefit from using the smaller subsets for more quickly finding settings that work well on the larger subsets. It would be interesting to study this to see if it works for Quick-Tune and how large speedups can be obtained.
* How well does the method work on other downstream tasks or data modalities?

---

> ### Author Response · Authors · 2023-11-20
> **Response to Reviewer omMr**
>
> Dear reviewer,
>
> thank you for the feedback. We address your questions as follows:
>
> 1. **The paper only considers computer vision classification, while related works such as LogMe (You et al., ICML’21) consider various downstream tasks (classification and regression), and modalities (vision and language). It can be impractically expensive though to evaluate the approach on also these other settings. How well does the method work on other downstream tasks or data modalities?**
>
> The reason that our method is focused on computer vision classification is that there already exists a well-established model hub in computer vision that is used by the community. However, we see no reason why our method would not work on other data modalities. We agree with the reviewer that it is infeasible to run on other data modalities at this stage and we believe this is part of future work.
>
> 2. **There are micro, mini and extended versions of the Meta-Album datasets, which poses the question if Quick-Tune could benefit from using the smaller subsets for more quickly finding settings that work well on the larger subsets. It would be interesting to study this to see if it works for Quick-Tune and how large speedups can be obtained.**
>
> We thank the reviewer for raising an interesting point. We would like to mention that this idea has already been investigated and verified by previous work [1]. As the number of epochs is the most common fidelity used in the domain, we focused on that. However, the fidelity in itself is a proxy and it could as well be dataset size.
>
> [1] Klein, A., Falkner, S., Bartels, S., Hennig, P., & Hutter, F. (2017, April). Fast bayesian optimization of machine learning hyperparameters on large datasets. In Artificial intelligence and statistics (pp. 528-536). PMLR.

---

> > ### Comment · Reviewer_omMr · 2023-11-20
> > **Thank you**
> >
> > Thank you for the response to my questions.

---

### Official Review · Reviewer_xZL3 · 2023-10-31

**Soundness:** 3 good
**Presentation:** 3 good
**Contribution:** 3 good
**Rating:** 8
**Confidence:** 3

**Summary:**

This work presents and evaluates an up-to-date combined model and hyperparameter selection method for transfer-learning/finetune setup. It works based on observing the learning curves of training runs and iteratively select which run to continue further based on bayesian optimization. The model maintains a performance predictor and a cost predictor, which is being updated as the search proceeds. The model parameters are/can be meta-learned.

The method is evaluated on a search space composed of 24 models on a pareto curve of number of parameters and ImageNet accuracy and on 86 tasks. The hyper parameter search space includes relevant settingslike finetune strategies, regularisation, data augmentation, optimizers, learning rate schedule.

The method is compared against other approaches such as: single model + HPO and model selection + HPO for selected model.

**Strengths:**

Model selection and hyperparameter optimization are practical problems many ML practitioners encounter. It is welcome to see a method able to tackle both together and that doing so provides benefits from doing it step wise. This reviewer is not aware of model selection papers / transferability estimation doing joint optimization in a recent finetune/transfer setting as in here.

The paper setting with 24 models and 86 datasets from Meta-Album when training for up to 1,4,16 hours seems reasonable and one practitioners can relate to.

**Weaknesses:**

I do not follow how the normalised regret is calculated. In particular how is y_max and y_min calculated? Is it provided by Meta-Album datasets? Is it the min/max of all runs ever done on this study? How significant is a 10% regret and is there any more expensive way to close that gap when using this approach?

The curves on plots like figure 3, shows a different behaviour between the methods. It is hard to predict if quick-tune always beats them or if that story changes as the wallclock time gets extended. It would be interesting to see if the search approaches regret 0 or stays ~10% above it.

One issue with model selection in particular with small datasets is overfitting, including to the validation set. I expected some discussion around this and also an explicit reference to which splits are used during which phase.

**Questions:**

What splits are being used to train, guide search and report test performance? It would be good to have that explicit in the text.

How is normalised regret calculated?

How do the curves in figure 3 look like when the wallclock time gets extended?

---

> ### Author Response · Authors · 2023-11-20
> **Response to Reviewer xZL3**
>
> Dear reviewer,
>
> thank you for your feedback. We address your questions below:
>
> 1. **I do not follow how the normalised regret is calculated. In particular how is y_max and y_min calculated? Is it provided by Meta-Album datasets? Is it the min/max of all runs ever done on this study? How significant is a 10% regret and is there any more expensive way to close that gap when using this approach?**
>
> We would like to point the reviewer to Appendix A1, where we describe how the normalised regret is calculated. The reviewer is correct in understanding that $y_{max}$ and $y_{min}$ are provided by the Meta-Album datasets. If the values were not provided, one could calculate the values from the runs done on the study, as the reviewer has intuitively noted. In both scenarios, we can get access to $y_{max}$ and $y_{min}$.
>
> A 10% regret means the configuration has a distance of 0.1 to the oracle configuration, based on the range of the task.
>
> As an example on why we use normalized regret,  suppose we are given two tasks A and B, and a method $f$:
>
> Let us assume we are using error rate, and we have the worst value in A of 0.75 and the best value 0.7. Let us assume our method $f$ finds hyperparameter configuration $\lambda$ during optimization that has a value of 0.725.
>
> Now suppose that for task B, we have a worst value of 1 and a best value of 0.5. Assume that during optimization our method $f$ finds a configuration with a value of 0.6.
>
> In task A, our unnormalized regret is 0.025, while in task B, our unnormalized regret is 0.1. However, observing the scale, our performance is located in the middle of the range of performances for task A. While, for task B, our performance is located in the 0.2 quantile. To overcome the aforementioned scenario, we use the normalization.
>
> Lastly, we would like to point out that reporting the normalized performance is a common practice in the domain [1][2][3].
>
> [1] Kadra, A., Janowski, M., Wistuba, M., & Grabocka, J. (2023, November). Scaling Laws for Hyperparameter Optimization. In Thirty-seventh Conference on Neural Information Processing Systems.
>
> [2] Mallik, N., Bergman, E., Hvarfner, C., Stoll, D., Janowski, M., Lindauer, M., Nardi, L., & Hutter, F. (2023). PriorBand: Practical Hyperparameter Optimization in the Age of Deep Learning. In Thirty-seventh Conference on Neural Information Processing Systems.
>
> [3] Chen, Y., Song, X., Lee, C., Wang, Z., Zhang, R., Dohan, D., ... & de Freitas, N. (2022). Towards learning universal hyperparameter optimizers with transformers. Advances in Neural Information Processing Systems, 35, 32053-32068.
>
>
> 2. **The curves on plots like figure 3, shows a different behaviour between the methods. It is hard to predict if quick-tune always beats them or if that story changes as the wallclock time gets extended. It would be interesting to see if the search approaches regret 0 or stays ~10% above it. How do the curves in figure 3 look like when the wallclock time gets extended?**
>
> We would like to point the reviewer to Figure 8 in the Appendix, which provides results for QuickTune on an extended runtime on all benchmarks. As can be observed by the results, the regret does not plateau at ~10% but keeps improving with time and it approaches 0. Additionally, we believe that Quicktune not only achieves a better final performance compared to the other methods, but it also achieves better anytime performance which is important for practitioners.
>
> 3. **One issue with model selection in particular with small datasets is overfitting, including to the validation set. I expected some discussion around this and also an explicit reference to which splits are used during which phase.**
>
>
> We would like to thank the reviewer for raising an interesting point. We used the same data set as the original meta-album implementation, which applies 20 % of the data for validation and 20% of the data for test while keeping the rest for training. We will update our manuscript to improve clarity for the camera-ready version as suggested by the reviewer. Regarding overfitting, our search space features varying fine-tuning strategies that are used to combat overfitting such as SP- regularization, Delta Regularization, and BSS-regularization. Moreover, we apply different data augmentation methods (mixup, cutmix) and various regularization techniques.
>
> For a more detailed description of our search space, we refer the reviewer to Section B.2 in the appendix.

---

### Official Review · Reviewer_LquR · 2023-10-31

**Soundness:** 3 good
**Presentation:** 4 excellent
**Contribution:** 4 excellent
**Rating:** 8
**Confidence:** 3

**Summary:**

* A challenge when in the  pretraining (PT) / finetuning (FT) paradigm is deciding (1) what PT model to use for a given task and (2) what hyperparameters to use when FT it. This paper presents a method to identify the best pretrained model and finetuning hyperparams to use for a new dataset, using bayesian optimization/
* The proposed method first pretrains surrogate models on a large meta-dataset of finetuning pipeline runs, which captures variation in datasets, model architectures.
* These surrogates are then used to define an acquisition function that defines how to select a finetuning pipeline (model specification and hyperparameter set) during each step of Bayesian optimization. Once more data is acquired, the surrogates are also updated.
* The specific acquisition function is a variation of expected improvement, including a term that captures the cost of running the finetuning pipeline (as opposed to being based purely on performance alone).
* In experiments, the surrogates are trained on a large set of learning curves from the timm benchmark. The algorithm is then applied to the Meta-Album dataset, and the results demonstrate that the proposed method outperforms baselines.
* Ablations where the meta-training step and cost-aware acquisition are removed demonstrate that both parts are important.

**Strengths:**

* Interesting problem choice, and original direction (to the best of my knowledge)
* Presentation, technical detail, and experiments are good quality.
* Proposed method has strong performance  on benchmarks considered
* Ablations of the cost-aware component of the acquisition function and meta-training were informative.

**Weaknesses:**

Paper is strong and has thorough results. The one thing I was curious about was how the method performs on other standard benchmarks such as Imagenet, and whether any of these results can be validation on different domains (eg text datasets, where finetuning is also very common).

**Questions:**

See above.

---

> ### Author Response · Authors · 2023-11-20
> **Response to Reviewer LquR**
>
> Dear reviewer,
>
> thank you for the feedback. We would like to clarify that the models considered in our search space, have been pretrained in ImageNet, which is why we do not include exactly ImageNet in our experiments. However, we would like to point the reviewer to Section C of the appendix, where we have done exactly what the reviewer suggests and we have evaluated how our method performs on other standard benchmarks such as Imagenette2-320 (a smaller version of ImageNet) and iNaturalist [2] against the baselines.
>
> We agree with the reviewer and we believe that our work opens up the chance to research further applications and model hubs. For instance, there is currently an explosion of new model hubs for LLMs in the community, such as OpenLLM, and we are looking forward to applying our methodology to this domain in future work.
>
> [1]https://www.tensorflow.org/datasets/catalog/imagenette
>
> [2] Van Horn, G., Mac Aodha, O., Song, Y., Cui, Y., Sun, C., Shepard, A., ... & Belongie, S. (2018). The inaturalist species classification and detection dataset. In Proceedings of the IEEE conference on computer vision and pattern recognition (pp. 8769-8778).

---

### Official Review · Reviewer_qaoF · 2023-11-07

**Soundness:** 3 good
**Presentation:** 3 good
**Contribution:** 2 fair
**Rating:** 8
**Confidence:** 4

**Summary:**

This paper introduces a Bayesian meta-learning approach called Quick-Tune to jointly optimize choice of models and hyperparameters for finetuning, i.e. a Combined Algorithm Selection and Hyperparameter Optimization (CASH) approach.  Quick-Tune builds on top a previous grey-box hyperparameter optimization approach called DyHPO with a cost-aware acquisition function and a meta-learning approach to warmstart loss and cost estimators.  Experiments on the Meta-Album benchmark for few-shot learning optimizing over a search space including models from TIMM shows Quick-Tune to efficiently select models and associated hyperparameters for new tasks, exceeding other standard hyperparameter methods as well as two-stage model selection and hyperparameter tuning baselines.  As part of training QuickTune, the authors also collect a meta-dataset of learning curves with associated hyperparameters and datasets to add to the set of meta-learning benchmarks.

**Strengths:**

- Paper is well written and clear.
- Quick-Tune addresses an important problem of how to efficiently select and tune models from a model hub for finetuning/transfer learning on a new dataset.
- Experiments appear to be thorough and high quality.

**Weaknesses:**

- In the ablation study, for the Micro and Mini cases, QT: -M, +C, +G (DyHPO with cost-awareness) performs as well as QT: +M, +C, +G (DyHPO with cost-awareness and meta-learning) which shows most of the benefit coming from the cost-aware aspect of QuickTune and not the meta-learning.
- Novelty is limited since the core of Quick-Tune is DyHPO, a prior HPO method.  Cost-aware acquisition functions have been used in the past and the approach of using meta-features and meta-learning good initializations for the estimators also lack originality.

**Questions:**

- What is the unnormalized performance of the approaches in Figure 3 and Figure 4?
- Instead of rank, how would Figure 1 look as a heatmap of unnormalized performance?

---

> ### Author Response · Authors · 2023-11-20
> **Response to Reviewer qaoF**
>
> Dear reviewer,
>
> thank you for your feedback and interesting insights. Below we answer your questions:
>
> 1. **In the ablation study, for the Micro and Mini cases, QT: -M, +C, +G (DyHPO with cost-awareness) performs as well as QT: +M, +C, +G (DyHPO with cost-awareness and meta-learning) which shows most of the benefit coming from the cost-aware aspect of QuickTune and not the meta-learning.**
>
>
> We thank the reviewer for raising an interesting point. We believe that Micro and Mini feature small tasks and as such, the time overhead of the initial evaluations does not penalize the performance during the optimization as much. In the case of more expensive tasks, this effect is reversed, and the meta-learning aspect helps improve performance by making use of prior information. We believe that meta-learning is an important aspect of our method, and in the case of small tasks it achieves similar performance to the non-meta-learned version.
>
> 2. **Novelty is limited since the core of Quick-Tune is DyHPO, a prior HPO method. Cost-aware acquisition functions have been used in the past and the approach of using meta-features and meta-learning good initializations for the estimators also lack originality.**
>
> Although our method extends previous work on gray-box optimization, we want to highlight that we make several novel contributions: 1) we show how to effectively combine meta-learning and cost-awareness in a gray-box setup, 2) we design a search space for finetuning based on a large model hub,  3) we introduce a meta-dataset for analysis and meta-learning, and 4) we show how to extend these ideas into the automated model selection and hyperparameter optimization for finetuning pretrained models.
>
> 3. **What is the unnormalized performance of the approaches in Figure 3 and Figure 4? Instead of rank, how would Figure 1 look as a heatmap of unnormalized performance?**
>
> We use the normalized performance since it makes the performance aggregation between the tasks correct (for a detailed explanation, we kindly point the reviewer to our response to reviewer xZL3. However, we share the plots requested by the reviewer with the unnormalized performance in the following link (anonymized repo):
>
> https://anonymous.4open.science/r/QuickTune-F637/figures/rebuttal_figures.md
>
> If the reviewer is satistifed with the clarifications and proposed changes we would appreciate a reflection of the discussion to the score. In case there are more questions, we are happy to answer them.

---

> > ### Comment · Reviewer_qaoF · 2023-11-23
> > **Post author response**
> >
> > Although I still believe the technical novelty is limited, I agree with the authors that the meta dataset is an important contribution and will raise my score from 6.

---

### Meta-Review · Area_Chair_ergm · 2023-11-27

**Metareview:**

The reviewers and meta reviewer all carefully checked and discussed the rebuttal. They thank the authors for their response and their efforts during the rebuttal phase.

The reviewers and meta reviewer all praised, among other things,
* The quality of the manuscript (clear and well-presented)
* The rigor and thoroughness of the experiments, with strong results of the proposed approach
* How important and central the tackled problem is (hyperparameter tuning with pretrained models), and how likely the paper will inform future research.

The paper is unanimously recommended for acceptance.

**Justification For Why Not Higher Score:**

N/A

**Justification For Why Not Lower Score:**

* High scores with strong alignment across reviewers
* Manuscript of an excellent quality (writing, clarity, structure)
* Thorough, rigorous and convincing experiments/ablation studies
* Proposed method has strong performance
* Central and topical problem
* The results will inform researchers and practitioners

---

### Decision · Program_Chairs · 2024-01-16

Accept (oral)